# Investigation of Geraniol Biotransformation by Commercial *Saccharomyces* Yeast Strains by Two Headspace Techniques: Solid-Phase Microextraction Gas Chromatography/Mass Spectrometry (SPME-GC/MS) and Proton Transfer Reaction-Time of Flight-Mass Spectrometry (PTR-ToF-MS)

Rebecca Roberts [1,2], Iuliia Khomenko [2], Graham T. Eyres [1], Phil Bremer [1], Patrick Silcock [1], Emanuela Betta [2] and Franco Biasioli [2,*]

1   Department of Food Science, University of Otago, P.O. Box 56, Dunedin 9054, New Zealand
2   Sensory Quality Unit, Research and Innovation Centre, Fondazione Edmund Mach, 38098 Trento, Italy
*   Correspondence: franco.biasioli@fmach.it

**Abstract:** Hop-derived volatile organic compounds (VOCs) and their transformation products significantly impact beer flavour and aroma. Geraniol, a key monoterpene alcohol in hops, has been reported to undergo yeast-modulated biotransformation into various terpenoids during fermentation, which impacts the citrus and floral aromas of the finished beer. This study monitored the evolution of geraniol and its transformation products throughout fermentation to provide insight into differences as a function of yeast species and strain. The headspace concentration of VOCs produced during fermentation in model wort was measured using Solid-Phase Microextraction Gas Chromatography/Mass Spectrometry (SPME-GC/MS) and Proton Transfer Reaction-Time of Flight-Mass Spectrometry (PTR-ToF-MS). In the absence of yeast, only geraniol was detected, and no terpenoid compounds were detected in geraniol-free ferments. During fermentation, the depletion of geraniol was closely followed by the detection of citronellol, citronellyl acetate and geranyl acetate. The concentration of the products and formation behaviour was yeast strain dependent. SPME-GC/MS provided confidence in compound identification. PTR-ToF-MS allowed online monitoring of these transformation products, showing when formation differed between *Saccharomyces cerevisiae* and *Saccharomyces pastorianus* yeasts. A better understanding of the ability of different yeast to biotransform hop terpenes will help brewers predict, control, and optimize the aroma of the finished beer.

**Keywords:** beer; fermentation; geraniol; biotransformation; SPME-GC/MS; PTR-ToF-MS

## 1. Introduction

Monoterpenoids are volatile organic compounds (VOCs) which can strongly impact food flavour. They are found in the essential oils of various plants, including hops [1,2]. Monoterpene alcohols, such as geraniol, linalool, nerol and α-terpineol, provide a "floral" or "citrus" aroma to beer. Interestingly monoterpenes such as citronellol and geranyl acetate have been detected in beer but not in hops [3]. While the generation of such compounds is not fully understood due to the complex chemical, physical and biochemical changes that occur throughout brewing and fermentation, *Saccharomyces* yeast have been reported to biotransform aroma compounds [4–7]. Specifically, geraniol has been reported to be a precursor for many of the monoterpenoids (via biotransformation) present in wine [8–10] or beer [4–6,11,12].

The complexity of hop essential oils and the various transformation reactions during fermentation make it challenging to determine the origin of VOCs produced in beer [3]. This challenge is particularly true for terpenoid compounds, which are responsible for much of the flavour and aroma in beer. Terpenoids can be present in the form of glycosides.

The cleavage of glycosides by yeast enzymes during fermentation can lead to the release of free terpenoids in beer, further contributing to the challenge of identifying the origin of aroma in beer [13]. Investigating the biotransformation of individual compounds in a model system could provide a better understanding of potential reaction pathways and the impact of different yeast strains on terpene production during fermentation. To date, research on biotransformation has mostly relied on techniques such as Solid Phase Micro Extraction (SPME) coupled with Gas Chromatography/Mass Spectrometry (GC/MS) to determine the VOC composition of beer. A limitation to this approach is that the analysis is typically performed only at the end of the fermentation, which does not provide all the information required to understand the dynamics of the biochemical reactions. Steyer et al. 2013, were among the first to evaluate the transformations of terpenes over time using Stir Bar Sorptive Extraction-Liquid Desorption (SBSE-LD) and GC/MS. The findings of their study indicated that geraniol underwent a transformation during fermentation by *S. cerevisiae*, resulting in the production of citronellol, linalool, nerol, citronellyl acetate, and geranyl acetate [14]. Alternative high throughput techniques, such as Proton Transfer Reaction-Time of Flight-Mass Spectrometry (PTR-ToF-MS) have been used to follow VOC development during fermentation [15]. PTR-ToF-MS provides dynamic measurements, increasing the understanding of volatile compounds' generation dynamics or reaction pathways, which can be useful for identifying the impact of different brewing conditions on the flavour and aroma [16].

Due to its high concentration in fresh hops, geraniol was an appropriate choice as the initial compound to investigate using model ferments. Previous studies have proposed several pathways for the biotransformation from geraniol. Still, limitations of these studies are the use of a complex starting material (whole hop cones) as well as the use of different microorganisms that are not commonly used in beer fermentation: *Cyanobacterium* [17], *Aspergillus niger* [18], *Castellaniella defragrans* and *Pseudomonas aeruginoa* [19]. An overview of the current literature related to beer and wine on the biotransformation of geraniol by *S. cerevisiae* is displayed in Figure 1 [5,6,20–22].

**Figure 1.** Previously identified biotransformation reactions of geraniol [5,6,20–22].

The current "gold standard" for the identification and off-line monitoring of VOCs is Gas Chromatography/Mass Spectrometry (GC/MS) which is a widely used analytical technique due to its high sensitivity, selectivity, wide linear range, versatility, and precision. GC/MS can detect VOCs at very low concentrations, even in the presence of other compounds. It can separate and identify individual VOCs in a sample, and it can

measure a wide range of VOC concentrations, from low parts-per-billion (ppb) to high parts-per-million (ppm) levels [23]. However, drawbacks of GC/MS include the long time required for a single analysis due to its labour-intensive sample preparation and quantitative analysis requiring reference standards [24]. In contrast, PTR-ToF-MS suits rapid quantitative analysis of VOCs in complex mixtures. It is an ultra-high sensitive technique that allows for the online analysis of VOCs based on their mass-to-charge ratio and can detect trace gases at ultra-low concentration levels (low ppt). A limitation of PTR-ToF-MS is that isomers are not distinguishable. Therefore, identification should be complemented by another analytical technique, such as GC/MS [25–27] or by implementing additional tools, such as fast-GC [28]. These two techniques have been previously used to measure VOCs in cheese, potatoes, infant formula, blueberries, milk, olive oil and truffles [26,29–31] and can support the identification of spectrometric peaks used for rapid monitoring over time [26,32]. The rapid PTR-MS-based methods also allow for the measurement of a larger number of replicates, making results statistically more robust. Therefore, two separate analyses were carried out at the same time in this study: one using SPME-GC/MS of a few select time points to identify the VOCs present, and the other using PTR-ToF-MS at more time points to monitor the generation of the VOC overtime more accurately.

The overall aim of the current study was to monitor the dynamic changes of geraniol during beer fermentation to understand and quantify in real time the point at which differences between *S. cerevisiae* and *S. pastorianus* yeast occurred and to provide new information on the biotransformation of geraniol during beer production.

## 2. Materials and Methods

### 2.1. Yeast Hydration and Model Wort Preparation

Commercially available yeast strains supplied by Fermentis (Lilles, France) were *Saccharomyces cerevisiae* strains SafAle US-05 and SafAle WB-06 and *Saccharomyces pastorianus* strains SafLager W-34/70 and SafLager S-23. Table 1 provides an overview of the samples measured with SPME-GC/MS and PTR-ToF-MS. The SPME-GC/MS samples were measured once every 24 h over a 5-day period, while the PTR-ToF-MS samples were measured once every 6 h over the same 5-day period. Each dried yeast strain was rehydrated separately in model wort. The model wort was prepared by dissolving 260 g of spray-dried malt extract (Briess Golden light) into 2 L deionized water (18 MΩ cm). For pH correction, 166 mg of calcium chloride ($CaCl_2$) was added [33]. In place of bittering hops, 76.7 mg of Iso-$\alpha$-acids (ICS—I4 Iso Standard; American Society of Brewing Chemists, St. Paul, MN, USA) was added to provide an international bitterness unit (IBU) of 20. The model wort was heated to 90 °C using a water bath and held for 10 min, then decreased to 20 °C using an ice bath. The main analytical characteristic of the model wort was pH 5.2, with a specific gravity of 12 °P. Each yeast strain starting density was $10 \times 10^7$ cells per mL, which was in line with manufacturing recommendations and best brewing practices.

**Table 1.** An overview of the samples measured with SPME-GC/MS and PTR-ToF-MS.

| Yeast Species | Yeast Strain | SPME-GC/MS | Measurement Frequency (h) | PTR-ToF-MS | Measurement Frequency (h) |
|---|---|---|---|---|---|
| *S. cerevisiae* | SafAle US-05 | ✓ | 24 | ✓ | 6 |
| *S. cerevisiae var. Diastaticus* | SafAleWB-06 | ✓ | 24 | ✓ | 6 |
| *S. pastorianus* | SafLager W-34/70 | - | - | ✓ | 6 |
| *S. pastorianus* | SafLager S-23 | - | - | ✓ | 6 |

### 2.2. Micro-Fermentations

Each 3 mL micro-fermentation consisted of model wort, yeast and 5 ppm of geraniol. In addition, samples without geraniol and samples without yeast served as blank controls. The samples were added into 20 mL glass head-space vials, sealed then placed into a ther-

mostatic autosampler tray (set to 20 °C) in a randomized order (CTC CombiPAL, CTC Analytics, Zwingen, Switzerland).

### 2.3. HS- SPME-GC/MS Analytical Conditions

VOCs were extracted using Head Space Solid Phase Microextraction on (HS-SPME-GC/MS) with 2-cm fibre coated with 50/30-μm divinyl benzene/carboxen/poly-dimethylsiloxane (DVB/CAR/PDMS, Supelco, Bellefonte, PA, USA). The fibre was exposed to the headspace for 40 min. The compounds absorbed on the SPME fibre were desorbed at 250 °C in the GC/MS injection port. The mass detector operated in electron ionization mode (EI, internal ionization source; 70 eV) with a scan range from $m/z$ 33 to 350. Analysis was carried out using Perkin Elmer Clarus 500 GC/MS equipped with an HP-INNOWax fused silica capillary column (30 m, 0.32-mm ID, 0.5-μm film thickness; Agilent Technologies, Palo Alto, CA, USA). The oven temperature was initially set at 40 °C for 1 min, then increased to 220 °C at 4 °C/min, increased to 250 °C at 15 °C/min and maintained for 2 min. Helium was used as carrier gas with a flow rate of 1.5 mL/min. Compound identification was based on mass spectra matching with NIST14/Wiley98 libraries. Linear retention indices were calculated under the same chromatographic conditions after the injection of a C7–C30 *n*-alkane series (Supelco).

### 2.4. PTR-ToF-MS Measurement

Headspace measurements were performed with a commercial PTR-ToF-MS 8000 apparatus from Ionicon Analytik GmbH (Innsbruck, Austria) in a standard configuration (V mode). The ionization conditions were as follows: 500 V drift voltage, 110 °C drift temperature, and 2.80 mbar drift pressure resulting in an E/N ratio of 130 Townsend (1 Td = $10^{-17}$ cm$^2$ V$^{-1}$ s$^{-1}$). Sample handling, headspace flushing and sampling were carried out using an autosampler (MPS MultiPurpose Sampler, Gerstel, Germany) specially adapted for PTR-ToF-MS [34]. The autosampler moved the sample from the incubation tray to the temperature-controlled purging site, connected to the PTR-ToF-MS inlet. Dynamic headspace analysis took place for 60 s with an acquisition rate of one mass spectrum per second between $m/z$ 15 and 349. Due to the high ethanol concentration, argon was added to the inlet system at a flow rate of 120 sccm, with the total flow rate of the system at 160 sccm. This prevented primary ion depletion and the formation of ethanol clusters that might affect the final quantification of volatiles [35]. The argon flow rate was controlled by a multi-gas controller (MKS Instruments, Inc., Andover, MA, USA). After measurement, the vial was moved back to the same position as the incubation tray, and the cycle was repeated on the following sample. During fermentation, the measurement was repeated every 6 h to monitor the fermentation process.

Deadtime correction, internal calibration of mass spectral data, and peak extraction were performed according to previously described procedures [36,37]. The peak intensity in ppb/v (parts per billion per volume) was estimated using the formula described in the literature [38]. The formula uses a constant value for the reaction rate coefficient (k = 2.10$^{-9}$ cm$^3$ s$^{-1}$).

Systematic errors can arise due to various factors, such as the use of a constant reaction coefficient, humidity, and fragmentation. However, in most cases, the error associated with measuring the absolute concentration of each compound is less than 30% and can be corrected post-analysis [36]. Certain mass peaks, such as those associated with isotopologues of $^{13}$C, $^{18}$O, and $^{27}$S, as well as water and ethanol clusters, were excluded from the dataset to minimise errors. Tentative compound identification was conducted by comparing the measured mass to the theoretical mass in the literature (Table 2). SPME-GC/MS was employed in conjunction with PTR-TOF-MS to confirm the tentative identification of compounds through the comparison of their mass spectra and chromatographic retention times.

**Table 2.** List of the peaks identified with PTR-ToF-MS. The measured mass, the identified mass, the sum formula and a tentative identification are given.

| Theoretical *m/z* | Measured *m/z* | Sum Formula | Chemical Class | Tentative Identification |
|---|---|---|---|---|
| 28.0062 | 28.006 | $C_2H_5^+$ | Alcohols | Ethanol Fragment |
| 33.0339 | 33.034 | $CH_4OH^+$ | Alcohols | Methanol |
| 48.0529 | 48.053 | $C_2H_5OH^+$ | Alcohols | Ethanol (isotopologue) |
| 59.0491 | 59.049 | $C_3H_6OH^+$ | Aldehydes/ketones | Propanol/acetone |
| 62.0317 | 62.031 | $C_2H_4O_2H^+$ | Esters and acids | Acetic acid |
| 64.0292 | 64.029 | $C_2H_6SH^+$ | Sulphur compounds | Dimethylsulfide |
| 69.0697 | 69.069 | $C_5H_8H^+$ | Terpene | Terpene fragment |
| 76.047 | 75.043 | $C_3H_6O_2H^+$ | Esters and acids | Propionic acid |
| 81.0699 | 81.07 | $C_6H_8H^+$ | Terpene | Terpene fragment |
| 83.0783 | 83.084 | $C_6H_{10}H^+$ | Terpene | Terpene fragment |
| 85.0654 | 85.064 | $C_5H_8OH^+$ | Aldehydes/Ketones | Pentanal/pentenone |
| 87.0439 | 87.043 | $C_4H_6O_2H^+$ | Ketones | Butanedione |
| 87.0803 | 87.08 | $C_5H_{10}OH^+$ | Alcohols | Pentanol |
| 94.0952 | 93.068 | $C_7H_7^+$ | Terpene | Terpene fragment |
| 95.0492 | 95.046 | $C_6H_6OH^+$ | Phenols | Phenol |
| 95.096 | 95.09 | $C_7H_{10}H^+$ | Terpenes | Terpene fragment |
| 97.0284 | 97.027 | $C_5H_4O_2H^+$ | Aldehydes | Furfural |
| 97.0642 | 97.057 | $C_6H_8OH^+$ | Aldehydes/Furans | Hexadienal/ethylfuran |
| 99.0802 | 99.079 | $C_6H_{10}OH^+$ | Aldehydes | Hexenal/methylpentenone |
| 101.0951 | 101.091 | $C_6H_{12}OH^+$ | Alcohols | Hexanol |
| 103.0749 | 103.074 | $C_5H_{10}O_2H^+$ | Esters and acids | Methylbutanoic acid |
| 107.0705 | 107.07 | $C_7H_6OH^+$ | Aldehydes | Benzaldehyde |
| 107.1071 | 107.102 | $C_8H_{10}H^+$ | Aromatic hydrocarbons | Xylene/ethylbenzene |
| 109.0712 | 109.059 | $C_7H_8OH^+$ | Phenols | Benzyl alcohol (cresol) |
| 111.0463 | 111.042 | $C_6H_6O_2H^+$ | Furans | Acetyl furan |
| 111.0804 | 111.076 | $C_7H_{10}OH^+$ | Aldehydes | Heptadienal |
| 113.0965 | 113.096 | $C_7H_{12}OH^+$ | Aldehydes | Heptanal |
| 115.1109 | 115.111 | $C_7H_{14}OH^+$ | Ketones | Heptanone |
| 121.0691 | 121.067 | $C_8H_8OH^+$ | Aldehydes | Methylbenzaldehyde-coumaran |
| 127.1117 | 127.112 | $C_8H_{14}OH^+$ | Ketones | Octenone/methylheptenone |
| 129.0911 | 129.091 | $C_7H_{12}O_2H^+$ | Esters and acids | Hexenyl formate |
| 129.1272 | 129.125 | $C_8H_{16}OH^+$ | Ketones | Octanone/Dimethylcyclohexanol |
| 131.1062 | 131.107 | $C_7H_{14}O_2H^+$ | Esters and acids | Heptanoic acid/hexyl formate |
| 135.1032 | 135.109 | $C_{10}H_{14}H^+$ | Aromatic hydrocarbons | Methylpropylbenzene |
| 136.1073 | 136.112 | $C_9H_{13}NH^+$ | Heterocyclic compounds | Butyl-pyridine/ethyl-propylpyridine |
| 137.132 | 137.133 | $C_{10}H_{16}H^+$ | Terpenes | Various monoterpenes |
| 141.1357 | 141.127 | $C_9H_{16}OH^+$ | Aldehydes | Nonanal |
| 143.1443 | 143.148 | $C_9H_{18}OH^+$ | Ketones/Aldehydes | Nonanone/nonanal |
| 151.1108 | 151.112 | $C_{10}H_{14}OH^+$ | Terpenes | Carvacrol/safranal |
| 153.0615 | 153.063 | $C_8H_8O_3H^+$ | Aldehydes | Vanillin, methyl salicylate |
| 153.1234 | 153.126 | $C_{10}H_{16}OH^+$ | Aldehydes | Citral |
| 155.1424 | 155.143 | $C_{10}H_{18}OH^+$ | Alcohols | Linalool/geraniol/a-terpineol/nerol |
| 157.1576 | 157.158 | $C_{10}H_{20}OH^+$ | Alcohols | Citronellol/dihydrolinalool |
| 171.1373 | 171.137 | $C_{10}H_{18}O_2H^+$ | Terpenes | Linalool oxide/Citronellic acid |
| 199.1677 | 199.169 | $C_{12}H_{23}O_2H^+$ | Terpenes | Citronellyl acetate |
| 201.1819 | 201.184 | $C_{12}H_{24}O_2H^+$ | Terpenes | Dihydrocitronellyl acetate |
| 205.1878 | 205.200 | $C_{12}H_{23}O_2H^+$ | Terpenes | Humulene |

Fragmentation Pattern Measurement

To improve the confidence in *m/z* used to monitor terpenes, the fragmentation patterns of pure standards were also measured (Table 3). Terpenoid standards; linalool, geraniol, α-terpineol, citral, citronellal, citronellal acetate, limonene, β-pinene, nerol, geranyl acetate, dihydrocitronellyl acetate, dihydrolinalool, myrcene, and caryophyllene, were diluted to a final concentration of 5 ppm through serial dilutions. These diluted standards were then analyzed with PTR-ToF-MS to obtain their fragmentation patterns. Preliminary experiments determined that the headspace concentrations were suitable and not below the detection limit. Compound identification was then carried out by comparing spectral data with fragmentation data. However, it is important

to mention the identification from PTR-ToF-MS remained tentative as isomeric product ions (both molecular ions and fragments) of different compounds can overlap at a given $m/z$.

**Table 3.** Terpenes, their molecular weight (MW), and formular and relative abundances of their different fragments determined by PTR-ToF-MS.

|  | MW (g/mol) | $m/z$ | 81 | 83 | 93 | 95 | 135 | 137 | 139 | 155 | 157 | 199 |
|---|---|---|---|---|---|---|---|---|---|---|---|---|
| Geraniol | 154.25 | $C_{10}H_{18}O$ | 82.03 |  | 100 | 51.25 |  | 46.06 |  | 0.27 |  |  |
| Citronellol | 156.27 | $C_{10}H_{20}O$ | 78.52 |  | 100 | 64.75 |  | 26.60 |  |  | 20.32 |  |
| Geranyl acetate | 196.29 | $C_{12}H_{22}O_2$ | 100 |  |  | 10.45 | 1.6 | 42.53 |  |  |  |  |
| Citronellyl acetate | 198.30 | $C_{12}H_{22}O_2$ | 40.47 | 100 |  | 11.10 |  |  | 41.32 |  |  | 25.51 |

*2.5. Data Analysis*

Table 1 provides an overview of the samples measured with SPME-GC/MS and PTR-ToF-MS. Multivariate statistical analysis was carried out using R 3.2.0 (R Foundation for Statistical Computing, Vienna, Austria) internal statistical functions and external packages, specifically: ggplot2 and ANOVA. Principal component analysis (PCA) was carried out using the R package "mixomics" [39] on the log-transformed and mean-centred data. A two-way ANOVA (yeast strain and time, $p < 0.001$) was used to determine the mass peaks with significant differences between yeast strains. When a monoisotopic mass peak was saturated, its isotopologue was considered a substitute ion.

**3. Results and Discussion**

*3.1. SPME-GC/MS Results*

The VOCs detected during the fermentation by the different yeasts with SPME-GC/MS are presented in Table 4. *S. cerevisiae* strains SafAle US-05 and SafAle WB-06 were selected to be measured using PTR-ToF-MS because previous experiments (data not shown) indicated that they were the most different in terms of their VOC composition. A preliminary data exploration was made using a principal component analysis (PCA), each point representing a distinct measurement (Figure 2). The first two principal components accounted for 88.44% of the total variability. Time-dependent evolutions are observed with different colours, representing the time points from day 1 to day 5. The first time point for all samples was similar, irrespective of yeast strain or compound, and clustered together close to the top left quadrant. After day 1, different evolutions were evident when comparing the samples with or without geraniol and separating by yeast strain. The loadings plot (Figure 2) shows the contribution of each VOC to the principal components at different time points and illustrates the differences in VOC evolution between yeast strains. On the other hand, the correlation circle plot (Figure 3) identifies which VOCs are most strongly associated with each principal component. In this study, terpenoid compounds such as geraniol (18), geranyl acetate (15), citronellol (16), and citronellyl acetate (13) played a crucial role in differentiating samples with added geraniol from the samples without added geraniol.

The time evolution of the detected terpenes (as the area under the curve) is displayed in Figure 4. In the samples with geraniol spiked, regardless of the yeast strain, the peak area of geraniol decreased over the first two days of fermentation and remained constant. This initial loss could result from several factors, including the removal of the compound from the solution due to $CO_2$ production by the yeast (stripping) during fermentation, as well as loss during sample measurement (purging). The ability of $CO_2$ to "blow off" the linalool during fermentation was investigated by Ferreira et al. (1996), who observed a reduction of 7.5% after 24 h [40].

When geraniol was not added, no terpenoids were detected. The terpenoids, geraniol, geranyl acetate, citronellol and citronellyl acetate were only detected in samples containing yeast to which geraniol had been added, with the concentration varying greatly depending on the yeast strain. The peak area of geranyl acetate on the second day of fermentation was more than three times higher with SafAle WB-06 than in the samples fermented with SafAle

US-05. Interestingly, geranyl acetate concentration in SafAle WB-06 decreased dramatically over time but remained relatively constant for SafAle US-05.

For the first two days of fermentation, citronellyl acetate was not detected in samples fermented with SafAle US-05. It then increased and plateaued until the final measurement on day 5. The formation of citronellyl acetate with SafAle WB-06 was comparable to the pattern for geranyl acetate; a dramatic increase followed by a decline. At the end of fermentation, the mean peak area of citronellyl acetate was similar for both yeast strains (SafAle US-05: $4.49 \times 10^6 \pm 5.53 \times 10^5$ and SafAle WB-06: $4.39 \times 10^6 \pm 9.80 \times 10^5$). The production of citronellol by SafAle US-05 and SafAle WB-06 followed a similar pattern, with similar abundance at each time point. Studies using GC/MS generally only provided a snapshot of the volatile profile of beer at a single time point, often at the end of fermentation. As a result, this approach has previously missed important dynamic changes in volatile production that occurred over the course of fermentation. Taking dynamic measurements throughout the fermentation provides a more comprehensive understanding of yeast metabolism and strain-dependent differences.

**Table 4.** Compounds detected at the end of fermentation with SPME-GC/MS.

| Number | Compound | Formula | CAS |
|---|---|---|---|
| 1 | Ethyl Acetate | $C_4H_8O_2$ | 141-78-6 |
| 2 | Ethanol | $C_2H_6O$ | 200-578-6 |
| 3 | Ethyl propanoate | $C_5H_{10}O_2$ | 105-37-3 |
| 4 | Ethyl butanoate | $C_6H_{12}O_2$ | 105-54-4 |
| 5 | Isobutyl alcohol | $C_4H_{10}O$ | 78-83-1 |
| 6 | Isoamyl acetate | $C_7H_{14}O_2$ | 123-92-2 |
| 7 | Isoamyl alcohol | $C_5H_{12}O$ | 123-51-3 |
| 8 | Ethyl hexanoate | $C_8H_{16}O_2$ | 123-66-0 |
| 9 | Ethyl octanoate | $C_{10}H_{20}O_2$ | 106-32-1 |
| 10 | Acetic acid | $CH_3COOH$ | 64-19-7 |
| 11 | Ethyl decanoate | $C_{12}H_{24}O_2$ | 110-38-3 |
| 12 | Isoamyl octanoate | $C_{13}H_{26}O_2$ | 2035-99-6 |
| 13 | Citronellyl acetate | $C_{12}H_{22}O_2$ | 150-84-5 |
| 14 | Ethyl 9-decenoate | $C_{12}H_{22}O_2$ | 67233-91-4 |
| 15 | Geranyl acetate | $C_{12}H_{20}O_2$ | 105-87-3 |
| 16 | Citronellol | $C_{10}H_{20}O$ | 106-22-9 |
| 17 | Ethyl dodecanoate | $C_{14}H_{28}O_2$ | 106-33-2 |
| 18 | Geraniol | $C_{10}H18O$ | 106-24-1 |
| 19 | Phenylethyl alcohol | $C_8H_{10}O$ | 60-12-8 |
| 20 | Octanoic acid | $C_8H_{16}O_2$ | 124-07-2 |

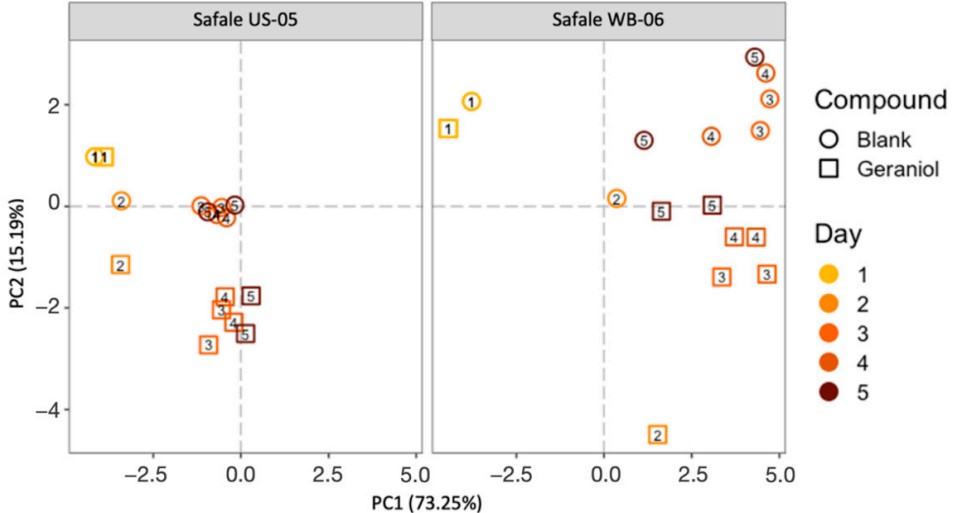

**Figure 2.** Score plot of the first two principal component analyses (PCA) of VOC produced during fermentation (5 days) by commercially available yeast: *S. cerevisiae* strain SafAle US-05 and *S. cerevisiae var Diastaticus* strain SafAle WB-06 (n = 2). Different colours represent the different days of fermentation.

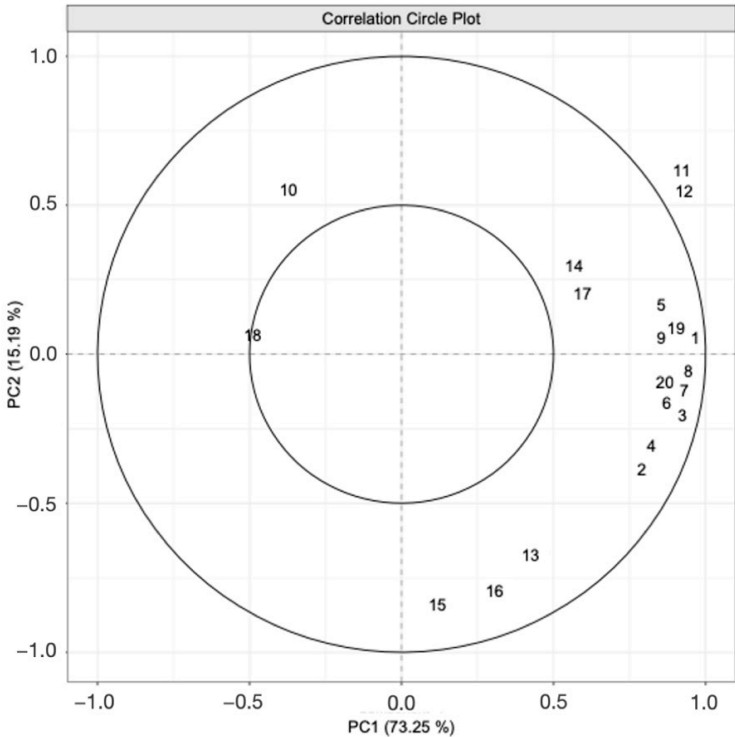

**Figure 3.** Correlation circle plots from the PCA applied to GC/MS data. Correlation circle plots display the correlation structure between compounds produced throughout fermentation in the space spanned by PC1 and PC2. The numbers represent the name of the compound displayed in Table 4.

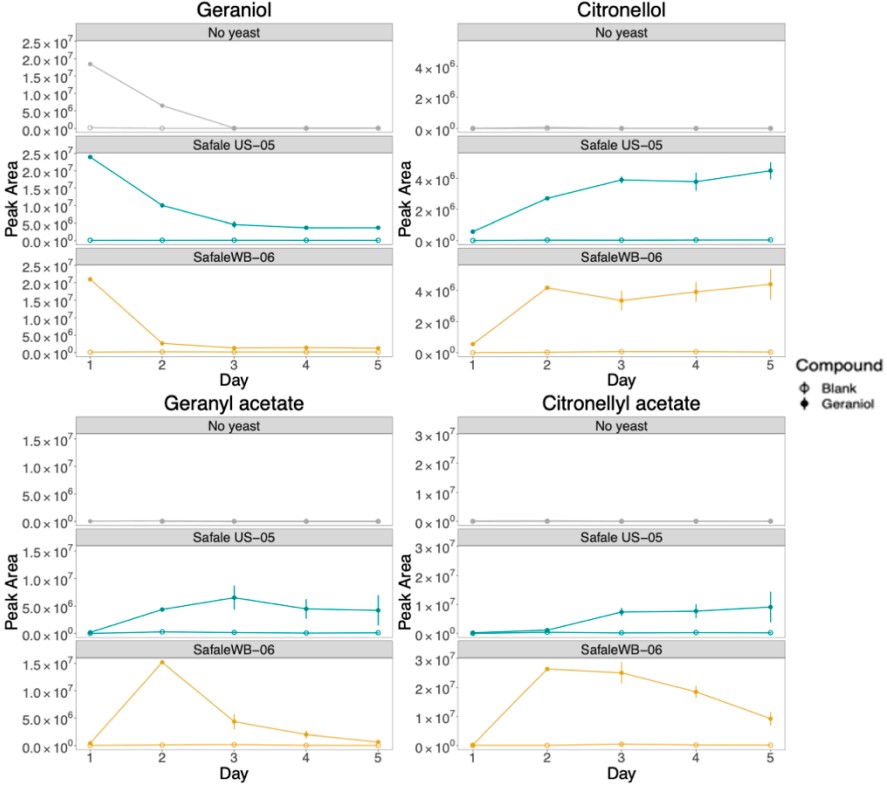

**Figure 4.** Time evolution of VOCs determined with SPME-GC/MS GC–MS. Quantification is given in the peak area detected under the curve, which can be quantitatively compared between the different measurements. Data presented as the mean peak area ± standard deviation of two replicates.

Steyer et al. (2013) [14] investigated terpene production between two yeast strains, *S. cerevisiae* strain S288c and a haploid strain 59a derived from a wine strain EC1118. Each yeast was added to a synthetic must medium (MS300) with geraniol (1 mg/L). Stir bar sorptive extraction (SBSE) and GC–MS analysis showed a rapid disappearance of geraniol from the medium, followed by the appearance of citronellol, nerol and linalool, and finally, geranyl- and citronellyl acetate were synthesized by both yeast stains. Neither nerol nor linalool was observed in the current study. Commercial yeast strains have been developed to meet the specific requirements of brewers with regard to stress resistance, brewing performance, enzyme release and the profile of aromatic compounds produced. Comparing the results from the current study highlights the impact of terpenoid addition and yeast on the compounds produced. Even yeast strains classified as the same species often show a high level of genetic divergence [41]. This genetic variability affects metabolism, and this information is not generally provided to brewers even though it could have a large impact on the VOC profile and flavour of the beer.

### 3.2. PTR-ToF-MS Results

*S. cerevisiae* strains SafAle US-05 and SafAle WB-06 and *S. pastorianus* strains SafLager W-34/70 and SafLager S-23 were selected for measurement with PTR-ToF-MS. Of the measured mass range between 15–245 *m/z*, 345 peaks were observed. After peak extraction and filtering, data calibration and filtration (eliminating isotopologues, water and ethanol clusters), 47 peaks were assigned to a sum formula and tentatively assigned to one or more compounds based on GC/MS identification and literature (Table 4). These tentatively identified compounds belonged to various chemical classes, many derived from yeast metabolism. PTR-ToF-MS measurement of four individual terpenoids produced fragment ions of masses 81 and 95 as non-isotopic ions (only $^{12}$C and $^{1}$H, not $^{13}$C or $^{2}$H) (Table 3). Mass 81 and 95 have previously been reported as terpene fragments [42–44]. Holzinger et al. (2000) proposed calculating the total monoterpene concentration from the signal of masses 67, 81, 95, 137 and 156 [42].

Development of Volatiles during Fermentation

The emission of ethanol and $CO_2$ during fermentation is directly associated with yeast activity as carbohydrates are converted into $CO_2$, ethanol, and hundreds of other secondary metabolites. Monitoring the evolution of ethanol (*m/z 47.049*) and carbon dioxide (*m/z 44.999*) was easily achieved as their protonated molecular ions are the predominant peaks [45]. No significant difference ($p < 0.001$) in the concentration of $CO_2$ between yeast strains was observed during fermentation (Figure 5). In contrast, the ethanol concentration in the samples produced by yeast strain WB-06 was significantly higher in the second, third and fourth measurement. There was no significant difference in the concentration of ethanol between yeast strains for the remainder of fermentation.

Secondary metabolites generated by yeast at the same time as $CO_2$ and ethanol are formed, which can influence the aroma and taste of beer. Variation in the metabolites across different yeast strains is what allows yeast to impart characteristic flavours to beer [46]. The selected 11 peaks were the protonated molecular ions of each terpenoid and their fragments identified from the SPME-GC/MS data: geraniol (*m/z 155.143, 137.132, 95.089, 93.952, 81.073*), geraniol acetate (*m/z 135.109*), citronellol (*m/z 157.158*), citronellyl acetate (*m/z 199.169, 139.141, 83.084*) and a fragment which is used to demonstrate total terpene concentration (*m/z 67.056*) as mentioned by Holzinger et al. (2000). The identification of the compound by *m/z* was done using standards, literature (if available) and by comparison to SPME-GC/MS data. Figure 6 displays the concentration (ppbV) of geranyl acetate (m/z *135.109*), geraniol (*m/z 155.143)*, citronellol (*m/z 157.158*) and citronellyl acetate (*m/z 199.169*) and measured by PTR-ToF-MS for four commercial yeast strains. The remaining 7 peaks (*m/z 139.141, 137.132, 95.089, 93.952, 83.084 and 81.073*) are displayed in the supplementary material.

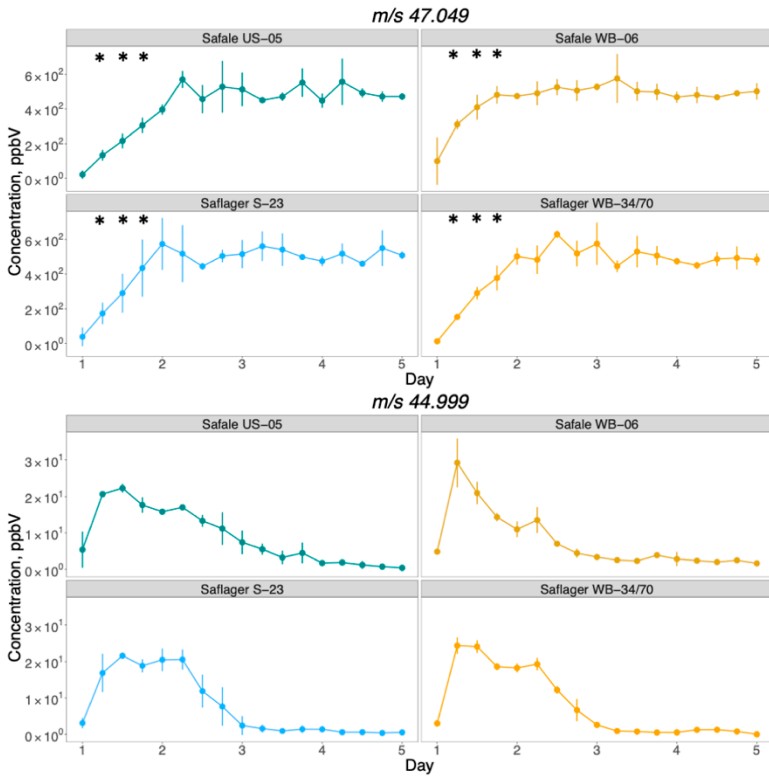

**Figure 5.** Mean concentration (ppbV) of ethanol (*m/z 47.049*) and carbon dioxide (*m/z 44.999*) during fermentation by commercially available yeast: *Saccharomyces cerevisiae* strain SafAle US-05, *Saccharomyces cerevisiae var Diastaticus* strain SafAle WB-06 and *Saccharomyces pastorianus* strains SafLager S-23 and SafLager W-34/70. Data presented as mean $\pm$ standard deviation of seven independent measurements. Asterisk (*) reflects statistically significant differences between strains with a *p*-value < 0.001, one-way ANOVA.

The signal evolution pattern for *m/z 157.158* is associated with citronellol as confirmed with pure standards and literature [47]. The gradual increase in citronellol over time was comparable to the results of SPME-GC/MS (Figure 4). The rapid analysis of samples with PTR-ToF-MS enabled the inclusion of additional yeast strains, such as *S. pastorianus* strains SafLager S-23 and SafLager WB-34/70, in the analysis. This increased sample throughput enabled the identification of species-specific differences that had not previously been observed. A significant increase in citronellol around the last two days of fermentation was identified, with the final concentration being highest in samples produced by *S. pastorianus* yeast. Species-dependent differences in the final concentration of citronellol from geraniol have been previously reported by Haslbeck et al. (2018). Unhopped wort with 70 µg/L of geraniol produced between 0.7–0.9 µg/L and 0.4 ug/L of citronellol by *S. cerevisiae* and *S. pastorianus*, respectively [48]. The old yellow enzyme (OYE) has been postulated as the enzyme responsible for this reduction of geraniol into citronellol [14]. The authors demonstrated this by fermenting using strains with the OYE2 gene either overexpressed or deleted. Deletion of the gene resulted in considerably less citronellol, and overexpression of the gene resulted in considerably more citronellol. The reduction of geraniol to citronellol may change the floral character of the beer to a more citrus-like character [4].

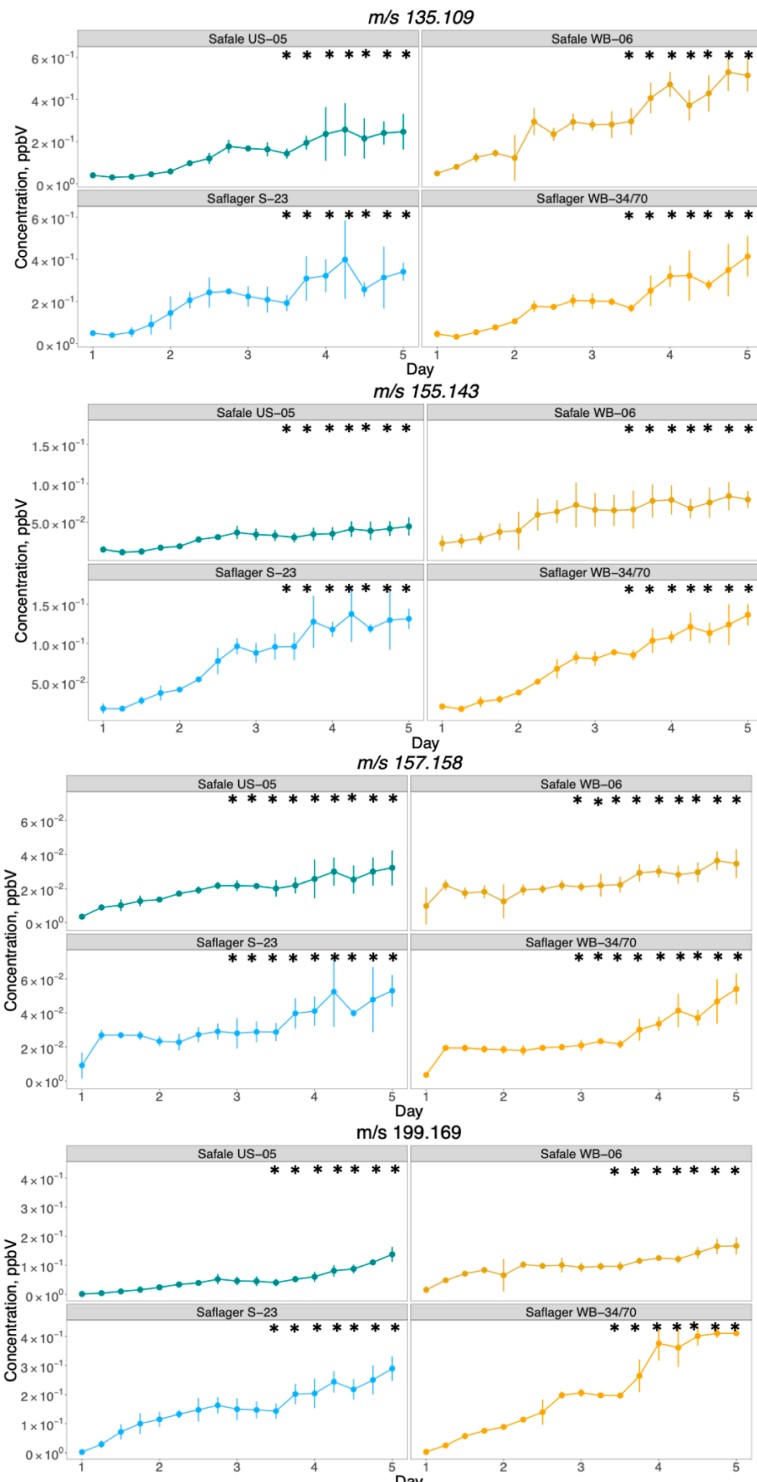

**Figure 6.** Mean concentration (ppbV) of geranyl acetate (*m/z 135.109*), geraniol (*m/z 155.143*), citronellol (*m/z 157.158*) and citronellyl acetate (*m/z 199.169*) during fermentation by commercially available yeast: *Saccharomyces cerevisiae* strain SafAle US-05, *Saccharomyces cerevisiae var Diastaticus* strain SafAle WB-06 and *Saccharomyces pastorianus* strains SafLager S-23 and SafLager W-34/70. Data presented as mean ± standard deviation of seven independent measurements. Asterisk (*) reflects statistically significant differences between strains with a *p*-value < 0.001, one-way ANOVA.

The unique signals *of m/z 83.084, m/z 139.141* and *m/z 199.169* were associated with citronellyl acetate. Monitoring the protonated molecule (*m/z 199.169*) showed increased concentration throughout fermentation by all yeast strains. Consistent with the SPMS GC/MS results,

an initial lag in the concentration of citronellyl acetate by SafAle US-05 was observed. All four yeast strains produced similar concentrations throughout fermentation until the end of day four, where a significantly higher concentration was produced by both *S. cerevisiae* strains. The unique signal of *m/z 135.109* was associated with geranyl acetate, which was determined using a pure reference standard. A significant difference in the concentration began halfway through day 3 and continued until the end of fermentation, with the highest concentration from samples produced by SafAle WB-06. The lowest concentration was from samples produced by SafAle US-05. SafLager S-23 and W 34/70 had a similar formation pattern throughout fermentation. Alcohol acetyltransferase (ATF) is the enzyme involved in the esterification of geraniol to citronellyl- and geranyl acetate [49,50]. Steyer et al. (2013) used a BY4741 strain with ATF1, or ATF2 genes deleted derived from *S. cerevisiae* S288C to display the involvement of ATF1 and ATF2 in the formation of terpenyl acetates through overexpression and deletion. When ATF1 and ATF2 were deleted, there was a drastic reduction in the formation of geranyl- and citronellyl acetate from geraniol. When the same gene was overexpressed, an increase in formation was observed. In the current study, the level of OYE and ATF expression in the different strains is likely to explain the differences observed. The loss of geraniol by biotransformation is mainly due to its reduction to citronellol catalyzed by OYE and acetylation to citronellyl acetate and geranyl acetate catalyzed by ATF. Brewers could utilize this knowledge to choose yeast strains that express these genes at desired levels, thereby achieving the desired concentrations of citronellol, citronellyl acetate and geranyl acetate in their beer.

### 3.3. Comparison between PTR-ToF-MS and GC/MS to Monitor the Formation of Compounds throughout Beer Fermentation

PTR-ToF-MS can measure more time points compared to GC/MS (four times a day vs. once a day for each micro-fermentation in our case). This is important, especially at the beginning of the fermentation, when many changes in VOCs are occurring, as evident in the initial 24 h for *m/z 157.158* (citronellol), *m/z 199.169* (citronellyl acetate) and *m/z 135.109* (geranyl acetate). An initial rapid increase in the concentration of citronellol was observed after 6 h. The concentration of citronellyl acetate started to increase, and finally, after 18 h, the concentration of geranyl acetate increased. A summary of this formation is shown in Figure 7. The delay of acetylation is likely due to the repression of ATF gene expression when oxygen (dissolved in the wort) is present [51]. The concentration of oxygen in wort gradually decreases during the first hours of fermentation. By 210 min, complete oxygen depletion is typically observed [52].

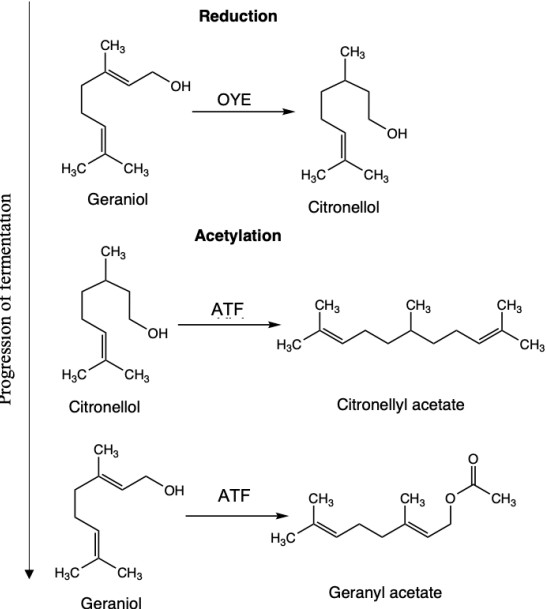

**Figure 7.** Proposed formation of citronellol, citronellyl acetate and geranyl acetate during fermentation. Enzymes (OYE and ATF) are based on literature [14,50,51].

## 4. Conclusions

The fate of geraniol during beer fermentation by *S. cerevisiae* or *S. pastorianus* was investigated as an example of biotransformation of hop flavour compounds occurring during beer fermentation. The reduction in the concentration of geraniol was closely followed by the detection of citronellol, citronellyl acetate and geranyl acetate. The ability of yeast to transform geraniol into these compounds has previously been identified. However, this is the first study to monitor the changes in real-time by direct injection mass spectrometry. The use of PTR-ToF-MS enabled differences between yeast strains to be identified during fermentation with high temporal resolution. The implementation of two separate techniques allowed for the identification of compounds (GC/MS), online monitoring and quantitative determination (PTR-MS). The results show that the concentrations of terpenoids detected throughout fermentation were impacted by the yeast species and strain. In general, a higher concentration of the detected terpenoids was produced by *S. pastorianus*. An example of strain-dependent differences was shown on the initial day of fermentation, where the increase in the concentration of citronellyl acetate (*m/z 199.169*) was slower in *S. cerevisiae* SafAle US-05 and WB-06 when compared to *S. pastorianus* SafLager S-23 and W-34/74. Biotransformation terpenoids increased in concentration at a similar rate for *S. pastorianus* strains, whereas *S. cerevisiae* strains SafAle- US-05 and WB-06 differed greatly. The difference between yeasts may be due to the diverse level of OYE and ATF expression, impacting the concentration of citronellol, citronellyl acetate and geranyl acetate, respectively. The concentration of some of the VOCs detected in the micro-fermentations (3 mL) may have different magnitudes when compared to industrial-sized fermentation. Still, the yeast differences and proposed pathways are expected to be comparable. Using this developed method to investigate terpenoids important for beer aroma is essential and might be used to investigate the effect of biological and technological parameters. To expand the current understanding of aroma generation during fermentation, analyzing more terpenoids with a similar experimental design will provide more valuable information on yeast production and transformation reactions. There is also a need to analyze different yeast strains, which would give the brewers additional information to manage and change the aroma of the beer to meet consumers' preferences.

**Supplementary Materials:** The following supporting information can be downloaded at: https://www.mdpi.com/article/10.3390/fermentation9030294/s1, Figure S1: Mean concentration (ppbV) of *m/z 139.141, m/z 137.132, m/z 95.089, m/z 93.952, m/z 83.084 and m/z 81.073* during fermentation by commercially available yeast: *Saccharomyces cerevisiae* strain SafAle US-05, *Saccharomyces cerevisiae var Diastaticus* strain SafAle WB-06 and *Saccharomyces pastorianus* strains SafLager S-23 and SafLager W-34/70. Data presented as mean ± standard deviation of seven independent measurements.

**Author Contributions:** Conceptualization G.T.E., P.S., P.B. and F.B.; Methodology, R.R., G.T.E., P.S., P.B., I.K. and F.B.; Investigation, R.R.; Formal analysis, R.R., I.K. and E.B.; Data Curation, R.R., I.K. and E.B.; Writing—original draft, R.R.; Writing—review and editing, F.B., G.T.E., P.S., P.B. and I.K.; Supervision, G.T.E., P.S., P.B. and F.B. All authors have read and agreed to the published version of the manuscript.

**Funding:** This research was funded by Provincia Autonoma di Trento (ADP FEM 2022).

**Institutional Review Board Statement:** Not applicable.

**Informed Consent Statement:** Not applicable.

**Data Availability Statement:** The data presented in this study are available on request from the corresponding author.

**Acknowledgments:** Rebecca Roberts is grateful to receive funding from the University of Otago (Doctorial Scholarship) and Fondazione Edmund Mach.

**Conflicts of Interest:** The authors declare no conflict of interest.

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
