# Peer review of "Investigation of Geraniol Biotransformation by Commercial Saccharomyces Yeast Strains by Two Headspace Techniques: Solid-Phase Microextraction Gas Chromatography/Mass Spectrometry (SPME-GC/MS) and Proton Transfer Reaction-Time of Flight-Mass Spectrometry (PTR-ToF-MS)"

_fermentation, doi:10.3390/fermentation9030294_

Round 1
Reviewer 1 Report
Thank you very much for giving me the opportunity to review the manuscript "Investigation of Geraniol…" by two internationally renowned food and flavor expert groups.
Even after reading the manuscript several times in great detail and being extremely picky, I cannot find any faults in the experimental design, the data evaluation or the drawn conclusions. Moreover, the results are of high importance and will not only be interesting to the analytical community, but also to brewers.
Thus, I recommend it for being published as it is.
Although it is typically done during typesetting, here I list the typos I spotted in the manuscript:
l70: …ARE the use…
l78: …WHICH is A widely used…
l85: "increasing time" sounds incorrect.
l99: …over time…
l107: : instead of ;
l115: Abbreviation IBU should be explained.
l130: The fibre was exposed to the headspace.
l152: Th is obsolete.
l186: Grammar.
l208: separating.
l226: evolution … is or evolutions … are.
l228: dayS
l252: Only some studies?
l264: WERE observed.
l276: use m/z instead of Th.
Table 4: What is the unit of the intensities? Percent? The most abundant ion is 100%?
Figure 5+6: Increase the increment on the y-axis for better readability.
l307: …by generated by…
l347: According to IUPAC "protonated molecule" should be used instead of parent ion.
l374: "especially shown" is not a good expression.
l398: …a higher concentration…were…
Author Response
Thank you very much for your positive feedback and careful revision. I have amended the typos as per your suggestions.
Reviewer 2 Report
Literature studies were carried out. Similar studies on the transformation of geraniol using Saccharomyces cerevisiae, where the changes in the content of analytes over time, have been conducted. As a result of biotransformation, the same compounds were formed as in the publication submitted for review. Analytes were similarly determined using GC-MS (Damien Steyer at all, Genetic analysis of geraniol metabolism during fermentation, Food Microbiology 33 (2013) 228-234). However, the production of wine was investigated. I have not found a similar publication on Saccharomyces pastorianus. Generally, the undertaken research has elements of novelty, especially that more modern analytical techniques were used.
Additional comments.
1. Due to the convenient way of searching the literature by browsing through abstracts, the names "Saccharomyces cerevisiae" and "Saccharomyces pastorianus" must appear in the abstract.
2. Table 4 - the columns should contain values accurate to the same number of decimal places.
3. Figure 6 - why are m/z values given, not compound names?
Author Response
I have included the study by Steyer el al in the introduction:
Steyer et al 2013, was one of the first to evaluate the transformations of terpenes over time using Stir Bar Sorptive Extraction-Liquid Desorption (SBSE-LD) and GC–MS. The findings of this study indicated that geraniol underwent transformation during fermentation by S. cerevisiae, resulting in the production of citronellol, linalool, nerol, citronellyl acetate, and geranyl acetate.[14]
Additional comments.
- Due to the convenient way of searching the literature by browsing through abstracts, the names"Saccharomyces cerevisiae" and "Saccharomyces pastorianus" must appear in the abstract.
I have added this into the abstract.
- Table 4 - the columns should contain values accurate to the same number of decimal places.
I have modified the table to include 2dp.
- Figure 6 - why are m/z values given, not compound names?
I have changed this to include the compound name and m/z.
Reviewer 3 Report
Very interesting work. It is of high analytical quality. A wide range of compounds were analysed compared to other articles.
Please only improve the quality (mainly increase) of the graphs presented in Fig. 4, 5 and 6.
Author Response
Very interesting work. It is of high analytical quality. A wide range of compounds were analysed compared to other articles.
Please only improve the quality (mainly increase) of the graphs presented in Fig. 4, 5 and 6.
Thank you for your feedback , I have improved the quality of the graphs.
Reviewer 4 Report
27:"PTR-TOF-MS was demonstrated to be......in beer." This sentence does not fit the previous content. Whether to consider deleting?
127 :The “Measurement frequency(hours)”is not clearly marked in this paragraph。
220 :Is it more appropriate to change “Diffierent shadings” to “diffierent colors”?
Author Response
27:"PTR-TOF-MS was demonstrated to be......in beer." This sentence does not fit the previous content. Whether to consider deleting?
I agree and have deleted this sentence
127 :The “Measurement frequency(hours)”is not clearly marked in this paragraph。
I have added:
Table 1 provides an overview of the samples measured with SPME/GC−MS and PTR-ToF-MS. The SPME/GC−MS samples were measured once every 24 hours over a 5-day period, while the PTR-ToF-MS samples were measured once every 6 hours over the same 5-day period.
220 :Is it more appropriate to change “Diffierent shadings” to “diffierent colors”?
I have changed from shadings to colours. Thank you for your feedback